# Study on a New Static Sealing Method and Sealing Performance Evaluation Model for PEMFC

**Xiaoyu Huang** [1], **Jinghui Zhao** [2], **Yichun Wang** [1], **Yuchao Ke** [3] **and Zixi Wang** [4],*

1    School of Mechanical Engineering, Beijing Institute of Technology, Beijing 100081, China; 15901082386@163.com (X.H.); wych@bit.edu.cn (Y.W.)
2    School of Automotive Engineering, Tongji University, Shanghai 201804, China; zhaojinghui@tongji.edu.cn
3    Joint Research Center, Zhongding Sealing Parts Co., Ltd., Ningguo 242399, China; keyc@zhongdinggroup.com
4    Department of Mechanical Engineering, Tsinghua University, Beijing 100084, China
*    Correspondence: zxwang@tsinghua.edu.cn; Tel.: +86-010-62794845

**Abstract:** The long-term stability and durability of seals are critical for various instruments and types of equipment. For static sealing, an important sealing state, there are currently two representative sealing methods, namely, pre-compressing static sealing and adhesive static sealing. In this paper, the characteristics and shortcomings of these sealing methods are summarized. At present, some static sealing requirements are urgent and difficult. For example, the deterioration of the sealing performance is an important factor which limits the service life of proton exchange membrane fuel cells and redox flow batteries. Therefore, a new method of static sealing whose sealing materials are rubber elastomers is proposed, named alterable static sealing. Then, its sealing processes are proposed. Furthermore, the actual contact area ratio $r$ is used as the standard for sealability. Based on the mathematical model of pre-compressing static sealing, the influence of interface bonding was considered, and the mathematical model of alterable static sealing was established. Moreover, the compensatory effect of alterable static sealing on the static sealing capacity of rubber elastomers was proved.

**Keywords:** proton exchange membrane fuel cell (PEMFC); static sealing; rubber elastomer; stress relaxation; interface bonding





## 1. Introduction

During the use of various instruments and types of equipment, not only the leakage of gas, fluid and solid particles between adjacent interfaces but also the ingress of external impurities should be prevented. Therefore, the long-term stability and durability of seals are critical, yet often overlooked.

Static sealing is an important sealing state which is widely used. The leakage rate is one of the direct standards for sealability. A significant increase in the leakage rate has been considered to indicate the loss of reliability of seals [1]. In the field of static sealing, leakage from two sealed surfaces, which is called interface leakage, is the main reason for sealing failures; meanwhile, leakage is greatly affected by the aging of sealing materials. Interface leakage means that when two sealed surfaces are in contact, macroscopic waviness and microscopic surface roughness exist in the two visually well-fitted surfaces, meaning the very well fitted contact condition is absent. Therefore, there are leakage paths between two surfaces because of these untouched areas [2].

In order to reduce the leakage rate, it is necessary to block leakage channels as much as possible to prevent the sealed medium passing through leakage channels between sealed surfaces and achieve better sealability. Therefore, in current static sealing methods, reducing surface roughness and increasing the actual contact area of two sealed surfaces have been widely used to block leakage channels. For example, the surface roughness of flanges

has been reduced to decrease the probability of untouched rough peaks between sealed surfaces. In addition, due to their excellent elasticity and resilience, rubber elastomers have been widely used as sealing materials.

Many researchers have studied the relationship between the leakage rate and the actual contact state of sealed surfaces. Persson, B.N.J. proposed a seepage theory in which the relationship between leakage channels and the actual contact state was analyzed with an assumption that the contacted interface consisted of square lattices, as shown in Figure 1 [3]. Based on seepage theory, Shi, J.C. et al. presented a mathematical model between the actual contact area ratio $r$ (the ratio of the actual contact area to the nominal contact area) and the possibility of the existence of leakage channels ($P_f$) [4]. Therefore, the leakage rate could be represented by the actual contact area ratio $r$, in order to represent the sealability.

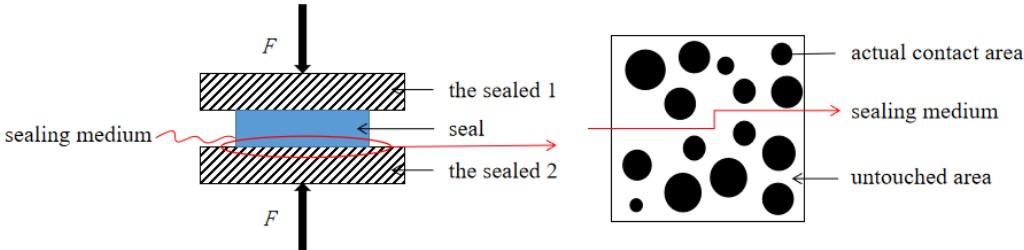

**Figure 1.** Schematics of seepage theory and interface leakage.

According to the methods of how to block leakage channels, there are two representative sealing methods of static sealing, which are summarized as follows.

### 1.1. Pre-Compressing Static Sealing

In pre-compressing static sealing, a seal made by rubber elastomers is placed between sealed surfaces. The sealed surfaces are pressed tightly by pre-tightening loads during operation. Compression deformation of elastomer seals can fill leakage channels and achieve the goal of sealing [5]. Pre-tightening loads should constantly be provided during operation to ensure the compression deformation of the elastomer seals, as shown in Figure 2. When the pre-tightening load increases to a certain value ($F_0$) [6], the leakage rate will basically remain stable, and the interface leakage can be blocked effectively, meaning a better sealing performance can be achieved [7]. However, the viscoelastic property of polymers can induce stress relaxation of seals under constant strain because of chain rearrangements, oxidation, chain scission and additional crosslinking, which lead to a loss of compression deformation over time. In other words, compression stress relaxation reflects the loss of sealing force over time. Therefore, if $F_0$ is used as the pre-tightening load, the compression deformation will not be enough after long-term use. In summary, the advantages of pre-compressing static sealing are mature processes, easy operation and easy disassembly. One of the disadvantages is that sufficient pre-tightening loads should be provided. Moreover, the lifetime of seals is limited by the viscoelastic property of polymers.

### 1.2. Adhesive Static Sealing

In adhesive static sealing, the liquid elastomer sealant is poured into the channels between the sealed surfaces. The liquid elastomer sealant has good fluidity and wettability before its vulcanization is completed, which allow the surface gaps to be fully filled. After vulcanization, the sealant and the sealed surfaces are bonded together, inseparably, and leakage paths are eliminated. The main applications of adhesive static sealing include form-in-place gaskets [8] and pouring sealants. In adhesive static sealing, bonding is formed between the liquid elastomer sealant through a crosslinking process, and the cohesive strength is produced, as shown in Figure 3a. If an adhesive is applied between the liquid elastomer sealant and the sealed surfaces, adsorption and mechanical interlocking are formed between the adhesive and the sealed surfaces through wetting effects. Meanwhile, bonding is formed between the adhesive and the liquid elastomer sealant through diffusion,

penetration and co-crosslinking processes. The cohesive strength is produced through internal curing reactions of the liquid elastomer sealant and adhesive [9], as shown in Figure 3b. Compared to pre-compressing static sealing, the sealing ability of adhesive static sealing does not depend on the compression deformation because of the liquidity of the liquid elastomer sealant. Thus, there are three main advantages of adhesive static sealing. First, the value of the pre-tightening load can be smaller. Second, the factors which affect compression deformation performance, such as stress relaxation, creep and elastic fatigue, can be ignored [10]. Third, the dimensional accuracy of the sealed surfaces can be lower. Thus, higher sealing reliability and a longer sealing life can be achieved. The disadvantage is that the processes are complex. Moreover, adhesive static sealing is a type of non-detachable sealing, which makes it difficult to adjust the relative positions during the assembling process.

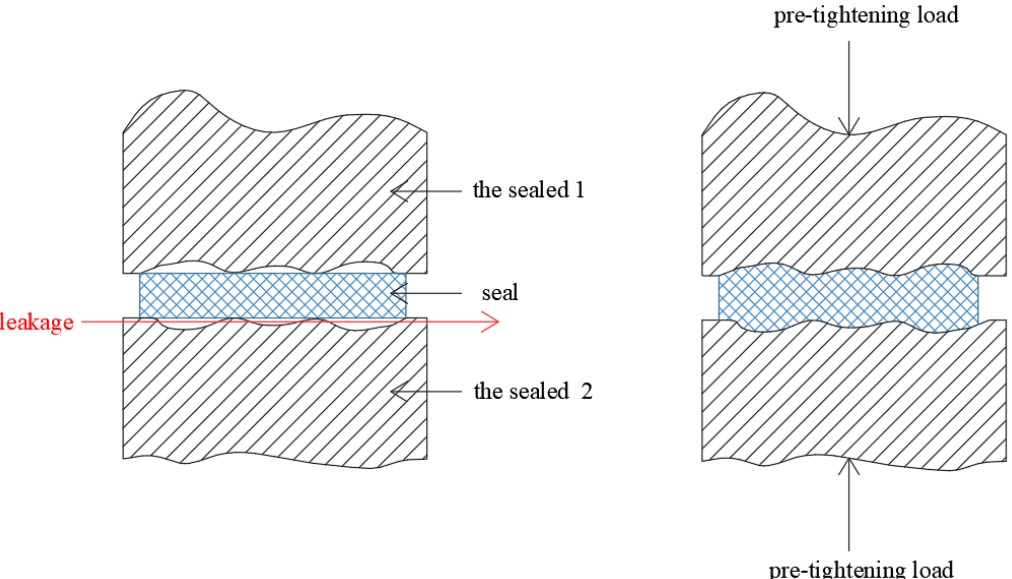

**Figure 2.** Schematics of the principle of pre-compressing static sealing.

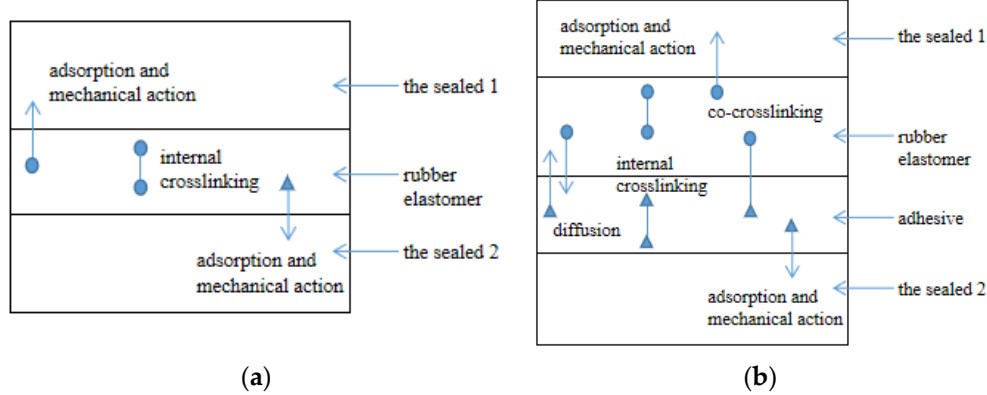

**Figure 3.** Schematics of the adhesion mechanism of: (**a**) adhesive static sealing without adhesive; (**b**) adhesive static sealing with adhesive.

Currently, an increasing amount of seals operate in severe and complicated environments. In order to find a typical complex sealing environment and explain its sealing requirements, we focused on batteries of electric vehicles. We found that the sealing requirement of proton exchange membrane fuel cells (PEMFCs) is a good example. PEMFCs have attracted increasing attention in recent years. Electric vehicles powered by PEMFCs have been commercialized in some countries, which is closely related to the goal of decarbonation.

In recent years, environmental and ecological problems have become increasingly serious, and the importance of reducing carbon emissions has been realized by many countries and regions. Some countries and organizations have proposed their decarbonation/decarbonization policies. For example, the goal of "achieving a carbon neutral and decarbonized society by 2050" was proposed by Japan at the National Congress in 2020 [11]. China stated its aim for "carbon neutrality by 2060" in the general debate of the 75th UN General Assembly in 2020 [12].

In the field of vehicles, electric vehicles perform better in reducing carbon emissions compared to traditional fuel vehicles. At present, the most widely used batteries in electric vehicles are lithium batteries. However, there are a large amount of flammable electrolytes in lithium batteries, and if thermal runaway occurs, there will be a high risk of flammability and explosion. Meanwhile, their low energy density, long charging time and short driving range have always been the disadvantages of lithium batteries. Fortunately, the fuels in PEMFCs, hydrogen and oxygen, operate with low carbon emissions. At the same time, the filling time of PEMFCs is similar to that of traditional fuel vehicles, but the driving range is longer [13]. Overall, besides durability, PEMFCs have greater advantages in reducing emissions and heavy metal pollution, meaning an increasing amount of electric vehicles are using PEMFCs.

A good sealing effect is the basic guarantee for the stable operation of PEMFCs. Once the sealing performance fails, very serious or even irreversible consequences would be caused.

Firstly, the leakage of hydrogen will waste hydrogen sources. Secondly, as we know, the lower explosion limit of hydrogen is low (4%), and the minimum ignition energy is very small (about 0.02 mj). As time goes on, the leaked hydrogen will accumulate in the volume space, meaning there is a higher risk of explosion [14]. Therefore, if the sealing performance is not good enough, the safety hazard caused by hydrogen leakage is greater than that in lithium batteries. Overall, the advantages of PEMFCs in electric vehicles can only be realized if the sealing performance is good.

Actually, sealing performance is a basic and important guarantee for not only PEMFCs but also other batteries used in electric vehicles, such as redox flow batteries (RFBs).

Since RFBs are also cost effective because of their ability to store a large amount of electrical energy in relatively efficient and inexpensive manners [15], both the European Commission and Department of Energy (DOE) are pushing RFBs to be employed in electrical vehicles [16,17]. RFBs are a type of power battery with a long cycle life and low carbon emissions. Meanwhile, the electrolyte solutions of RFBs are recyclable and mostly nonflammable, meaning fire risks can be decreased. When we compared PEMFCs with RFBs, we found that there are many similarities, which are as follows:

(1) In terms of principle, hydrogen and oxygen are fuels of positive and negative electrodes, respectively, and these fuels are stored in gas storage tanks. The positive and negative electrodes are separated by a proton exchange membrane. Under operating conditions, fuels are transported to electrodes, and then electric energy can be generated by electrochemical reactions. PEMFCs are a type of safe and durable power battery with clean fuels and low carbon emissions. In RFBs, positive and negative liquid electrolytes (containing metal ions) are stored in two electrolyte tanks which are arranged outside of the cells. The positive and negative electrodes are separated by an ion exchange membrane. Under operating conditions, liquid electrolytes are transported to electrodes [15].

(2) In terms of structure, the single cells of PEMFCs include a membrane electrode (MEA), bipolar plates and a sealing gasket, as shown in Figure 4a. The single cells of RFBs include an ion exchange membrane, electrodes, plates and a sealing gasket [18,19], as shown in Figure 4b [20]. These sealing gaskets can effectively prevent the leakage of fuel gases and liquid electrolytes.

(3) In terms of sealing difficulties, the difficulties of these two types of batteries are similar as well. There are four main issues. (1) The total length of the sealing line is long,

and the amount of sealing is large. (2) Sealing performance degrades over time with the operation of batteries as they are exposed to complex and corrosive working environments. (3) There are disassembly and reassembly requirements during the stacking process. (4) The lifetime of seals should be consistent with the service time of stacks. Therefore, more reliable sealability is necessary and urgent for the stable operation of PEMFCs and RFBs.

We will mainly take PEMFCs as an example to illustrate the sealing processes.

Up to now, for pre-compressing static sealing, which is widely used in the sealing of stacks, it is difficult to overcome the performance deterioration caused by the stress relaxation of rubber elastomers. The analysis reports of the U.S. Department of Energy state that the service time of PEMFCs should not be less than 5000 h under dynamic conditions [21,22]. However, research by Cui, T. et al. showed that, when stress relaxation was considered, the service life of seals made by liquid silicone rubber was about 5000 h in deionized water at 70 °C [23]. The lifetime of seals approached the lifetime limit of PEMFCs, even though the acidic environment was not considered in this research. Therefore, it can be seen that the need to improve the sealing performance of PEMFCs is urgent.

In addition, for adhesive static sealing in PEMFCs, the common operations are introduced as follows. The bipolar plates coated with an adhesive, MEA and an unvulcanized liquid elastomer sealant are placed in the mold. Then, the vulcanization of the liquid elastomer sealant, the curing of the adhesive and the interfacial reactions are completed simultaneously under heat and pressure. However, the single cells and stacks formed in this way cannot be disassembled, making it difficult to meet the disassembly and assembly requirements of PEMFCs during the stacking process. Moreover, the cost and the complexity of the processes are higher than those in pre-compressing static sealing.

Based on the disadvantages of the above sealing methods, a new sealing method named alterable static sealing is proposed in this paper. In the field of alterable static sealing, interface bonding between the solid polymer materials of the sealed surfaces could be formed under certain environmental stimuli, meaning that the attenuation of the sealing performance caused by the stress relaxation property of rubber elastomers could be compensated.

In the phrase "alterable static sealing", "alterable" is the limitation of our new static sealing method, which means that the source of the sealing force is not constant in the whole sealing cycle, making it different from traditional static sealing methods. Before applying the specific environmental stimulus, sealing forces are provided by pre-tightening loads and the resilience of rubber elastomers. If the pre-tightening loads are removed, the seals and the sealed surfaces can be easily separated. During the application stage of the specific environmental stimulus, the sealing forces are provided by pre-tightening loads, the resilience of rubber elastomers and the interfacial bonding strength. After the environmental stimulus is removed, if the pre-tightening loads are removed, the seals and the sealed surfaces could be integrated due to the bonding of the interface. The ratio of the interface bonding force to the total sealing force can be controlled through controlling the application method of the stimulus.

In this paper, we quantified the compensatory effect of interface bonding on the sealing performance. The actual contact area ratio $r$ was used as the evaluation index of the sealing performance. Importantly, considering the influence of interface bonding, the model of alterable static sealing was established based on the original mathematical model of pre-compressing static sealing. Additionally, the advantages of alterable static sealing were proven.

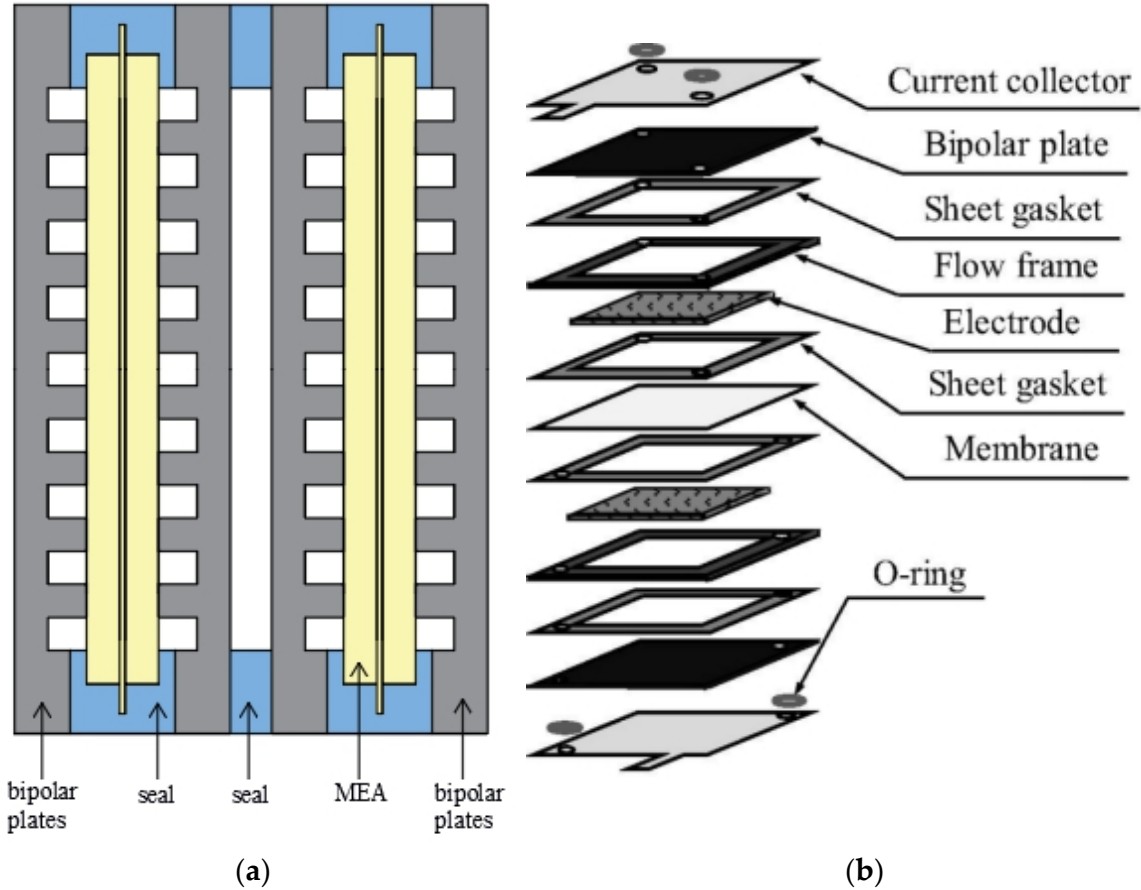

**Figure 4.** Schematics of the structures of: (**a**) proton exchange membrane fuel cells (PEMFCs); (**b**) redox flow batteries (RFBs) [20] (Copyright: © 2021 by Elsevier. Licensee Ms. Xiaoyu Huang. License Number 5181940503674. License date 4 November 2021).

Alterable static sealing, as introduced in this paper, can not only ensure detachability during the assembly process but also reduce the requirements of the pre-tightening loads, the aging speed of elastomers and the dimensional accuracy of the sealed surfaces. The interface bonding process can be adjusted as well.

## 2. Theoretical Models

In this paper, we propose the working characteristics of alterable static sealing. During the initial stage, two sealed surfaces are separated, and the pre-compressing static sealing stage is achieved by pre-tightening loads. Then, after the environmental stimulus is imposed, the solid polymer materials of the two sealed surfaces can gradually bond, causing the sealing stage to transfer into a new stage with interface bonding; this new sealing stage is named alterable static sealing.

### 2.1. The Sealing Processes of Alterable Static Sealing

We propose the scheme of the implemented sealing processes of alterable static sealing in this section. Firstly, we modified two sealed surfaces, one with polymer A and the other with polymer B (or modified two sealed surfaces and the rubber elastomer gasket between them with polymer A and polymer B). Two solid polymer layers were formed on the two sealed surfaces. Secondly, the pre-tightening load was applied to the sealed surfaces, and its value was smaller than that used in pre-compressing static sealing. Thirdly, an environmental stimulus was employed on the sealed surfaces. On the interface, the diffusion was intensified, and stable chemical bonds were formed between the terminal reactive groups of polymers A and B. Therefore, the interaction on the interface could be

enhanced, and bonding strength could be produced within the sealed elements, as shown in Figure 5.

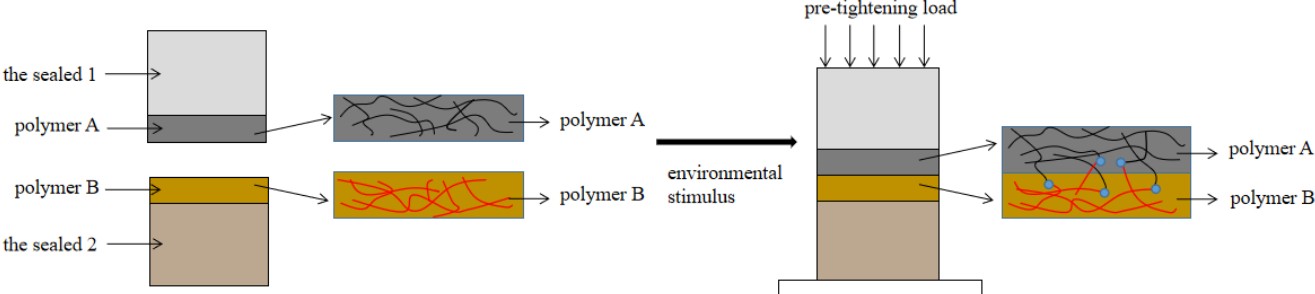

**Figure 5.** Schematics of implemented processes of alterable static sealing.

There are a variety of possible factors that can be used as the environmental stimulus, such as high temperature, light, the pH value, the electromagnetic field and some specific chemical reagents. The application time of stimulation is related to the thermal movement rate of chain segments and the chemical reaction kinetics. In laboratory tests, the type of stimulation depends on the bonding mechanism of solid polymers on two sealed surfaces. For example, Stukenbroker, T. et al. used high temperature (100 °C) as the stimulation, and the interfacial bonding of polydimethylsiloxane was realized, which depended on the reaction of the alkenylamide bond and amino groups [24]. Röttger, M. et al. considered the pH responsiveness of pentamembered cyclic borate. The interfacial bonding of some thermoplastics such as poly (methyl methacrylate), polystyrene and high-density polyethylene was realized in alkaline solutions [25]. Denissen, W. et al. realized the interfacial bonding of poly (vinylogous urethane) at high temperature (150 °C) based on the transamination of vinylogous urethanes [26]. Yan, P. et al. manipulated near-infrared light to realize the interfacial bonding of carbon nanotube-polyurethane nanocomposites, in which the exchange reaction of carbamates was promoted based on the photothermal conversion effect of carbon nanotubes [27].

A stable actual contact could be formed between the sealed surfaces because of interface bonding. Thus, the negative influence of stress relaxation on the leakage rate could be reduced. Detachability could be guaranteed during the assembly process by controlling the environmental stimulus conditions.

We took the static sealing of PEMFCs as an example to describe the scheme of the implemented sealing processes of alterable static sealing. The surfaces of bipolar plates and the sealing frame of MEA made by rubber elastomers were modified by polymers A and B, respectively, meaning two solid polymer layers were formed on the above surfaces. Then, a single cell was assembled, and the battery pack was obtained by stacking MEA of the next single cell on the bipolar plate of the previous single cell. A PEMFC stack could be obtained after the end plates of the stack were added at both ends of the battery pack and the stack was fastened by screws. After the stack was assembled and adjusted, the environmental stimulus could be applied. The segment diffusion of polymers A and B was intensified, and stable chemical bonds were formed between the terminal reaction groups of polymers A and B on the sealed interface. The bonding strength was generated to block leakage channels.

In the sealing of PEMFCs, as an application example of our sealing method, because of the service environment of the seals (acid, high humidity), high temperature is a suitable stimulation.

### 2.2. The Relationship between Sealing Performance and the Actual Contact Area Ratio r on Pre-Compressing Static Sealing

In order to describe the relationship between the stress relaxation and sealing performance of rubber elastomers, we first considered the relationship between the stress relaxation behavior of rubber elastomers and time $t$.

The rubber elastomers have the properties of both viscosity and elasticity. Therefore, the mathematical model for the viscoelastic behavior should include elements representing these properties. The most basic model is the Maxwell model. Based on the Maxwell model, the generalized Maxwell model has been proposed and applied to predict the stress relaxation behavior of rubber elastomers [28].

In this paper, the generalized Maxwell model was employed to describe the stress relaxation behavior [28], as shown in Figure 6.

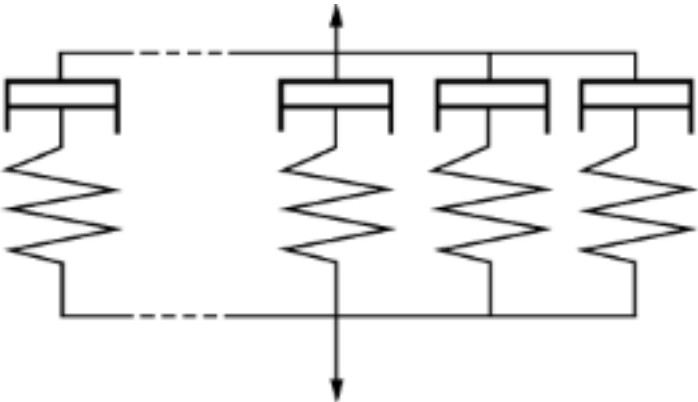

**Figure 6.** Schematic drawing of the generalized Maxwell model.

When the strain is a constant, the relationship between stress and time $t$ can be expressed as

$$\sigma(t) = \varepsilon_0 E_n + \varepsilon_0 \sum_{i=1}^{n-1} E_i e^{-t/\tau_i} \tag{1}$$

where $\sigma(t)$ is the stress at time $t$, $\varepsilon_0$ is the initial strain when $t = 0$, $E_n$ is the elastic modulus of the polymers when they are not affected by viscoelastic behaviors, and $\tau_i$ and $E_i$ are the relaxation time and the elastic modulus corresponding to the $i$th Maxwell model, respectively. Equation (1) is often called the Prony series, and researchers have proved that it is justified for the stress relaxation behavior of polymeric elastomers [29,30].

The value of the parameters in Equation (1) could be obtained from long-term stress relaxation experiments of rubber elastomers. Furthermore, based on time–temperature superposition (TTS), the experimental results found at high temperature could be used to predict long-term stress relaxation at low temperature [29].

A mathematical model between the actual contact area ratio $r$ and the existence probability $P_f$ of leakage channels was presented by Shi, J.C. et al., as shown in Equation (2). It was found that there is a threshold of leakage, $r_c$. When $r$ is bigger than $r_c$, $P_f$ is low at zero, and the sealing performance is excellent [4]. Moreover, Dapp, W.B. proposed that the value of $r_c$ is about 0.42 by numerical simulations [31]. Additionally, the relationship between the leakage state and $r$ has been widely investigated [3,32,33].

$$P_f(r) = 1 - \frac{1}{2}[1 + erf(\frac{r - r_c}{\sqrt{2}\sigma})] \tag{2}$$

Based on these studies, in this research, the actual contact area ratio $r$ was used to characterize the sealing performance, in order to analyze the influence of stress relaxation.

Next, the relationship between stress and the actual contact area ratio $r$ could be determined by in situ compression experiments. Pang, M.H. et al. observed the liquid

infiltration processes between rubber elastomers in situ and obtained the relationship between stress and the actual contact area ratio $r$, which shows a power function law [34]. The fitting relationship could be described as Equation (3).

$$\sigma(r) = A_r r^{B_r} \tag{3}$$

where $\sigma(r)$ is the stress, $r$ is the actual contact area ratio, and $A_r$ and $B_r$ are fitting parameters.

In the field of pre-compressing static sealing, the relationship between the actual contact area ratio $r$ and working time $t$ meets the requirement in Equation (4).

$$r = \left[ \left( \varepsilon_0 E_n + \varepsilon_0 \sum_{i=1}^{n-1} E_i e^{-t/\tau_i} \right) / A_r \right]^{1/B_r} \tag{4}$$

*2.3. The Bonding Model of Interpenetration Region around the Interface*

During the interface bonding processes of alterable static sealing, polymer segments diffuse, and chemical bonds and intermolecular forces are formed. In this section, a theoretical model is used to describe these processes.

2.3.1. Model of Covalent Polymer Networks

The polymer networks in alterable static sealing include the crosslinked polymer networks A and B, and the interpenetrating network at the interface. Here, we used the Arruda–Boyce model to model networks A and B. The Arruda–Boyce model assumes that the polymer chains have the same length and self-organize into body-centered eight-chain structures [35], as shown in Figure 7.

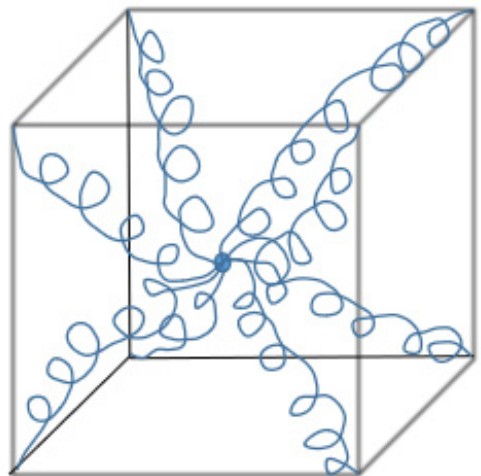

**Figure 7.** Schematic drawing of the Arruda–Boyce model.

The free energy density function of this deformed network is

$$W_0 = N_0 n_0 k_B T [\beta/\tanh\beta + \ln(\beta/\sinh\beta)] \tag{5}$$

where $\beta = L^{-1}(\Lambda/\sqrt{n_0})$, $L(x) = \coth(x) - 1/x$ is the Langevin function, $L^{-1}(x)$ is the inverse of the Langevin function, $\Lambda$ is the chain deformation, $k_B$ is the Boltzmann constant and $T$ is the absolute temperature in kelvin.

In order to contact the chain deformation to the macroscopic deformation, we used a deformation model, and the deformation of the chain is

$$\Lambda = \sqrt{I_1/3} \tag{6}$$

where $I_1$ is the first invariant of the right Cauchy green tensor [35].

We assumed that the volume of the material is a constant, and the degree of compression $\lambda$ is the ratio of the initial thickness and the final thickness in the main compression direction ($\lambda_1 = 1/\lambda$, $\lambda_2 = \lambda_3 = \sqrt{\lambda}$), meaning that the deformation of the chain is

$$\Lambda = \sqrt{I_1/3} = \sqrt{(2\lambda + \lambda^{-2})/3} \tag{7}$$

The Cauchy stress along with the $\lambda_1$ direction can be written as

$$\sigma_1 = N_0\sqrt{n_0}k_{\mathrm{B}}T\frac{(1-\lambda^{-3})}{\lambda\sqrt{6\lambda+3\lambda^{-2}}}L^{-1}\left(\sqrt{\frac{2\lambda+\lambda^{-2}}{3n_0}}\right) \tag{8}$$

Next, we established the model of the formation processes of the interpenetrating network between the interface. We assumed the chain lengths of polymers A and B are inhomogeneous. Polymers A and B are composed of m types of networks interpenetrating in the material bulk space. In each unit cube, the polymer chains can be established in the Arruda–Boyce model [36]. In this model, we ignored the effects of chain entanglement and chain crosslinking.

Each polymer chain is composed of freely connected Kuhn segments. The length of each segment is b, and the number of Kuhn segments of the *i*th chain is *ni* [37]. The numbers of *i*th polymer chains per unit volume of polymers A and B are $N_{i\mathrm{A}}$ and $N_{i\mathrm{B}}$, respectively, where *1 ≤ I ≤ m* and *n1 ≤ n2 ≤... ni... ≤ nm*, as shown in Figure 8. The statistical distribution of the number of chains is shown as Equation (9).

$$P_{i\mathrm{A}}(ni) = \frac{N_{i\mathrm{A}}}{\sum_{i=1}^{m}N_{i\mathrm{A}}}, \; P_{i\mathrm{B}}(ni) = \frac{N_{i\mathrm{B}}}{\sum_{i=1}^{m}N_{i\mathrm{B}}} \tag{9}$$

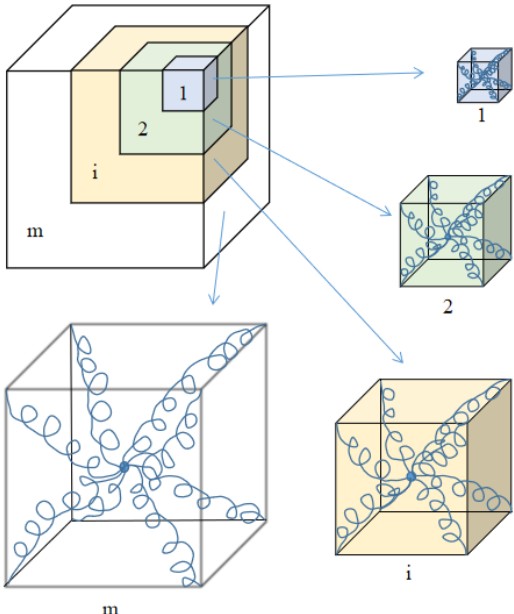

**Figure 8.** Schematics of an interpenetrating network model.

For the undeformed chain, the end-to-end distance of the *i*th network can be approximated as

$$r_i^0 = \sqrt{n_i}b \tag{10}$$

Under compression, the deformed end-to-end distance is $r_i$, with the corresponding chain compression as

$$\Lambda_i = r_i/r_i^0 \tag{11}$$

The free energy of the deformed $i$th chain is

$$w_i = n_i k_B T [\beta_i / \tanh\beta_i + \ln(\beta_i / \sinh\beta_i)] \tag{12}$$

where $\beta_i = L^{-1}(\Lambda_i / \sqrt{n_i})$ [35–37].

Additionally, the chain force on the deformed $i$th chain is defined as

$$f_i = \partial w_i / \partial r_i = k_B T \beta_i / b \tag{13}$$

### 2.3.2. Diffusion Model of Polymer Chains

As we assumed in Section 2.3.1, the $i$th chain self-organizes into a body-centered eight-chain structure. Two sealed surfaces are modified by polymers A and B. The movement of polymer segments in polymers A and B is increased after an environmental stimulus condition is applied. The terminal reactive groups on the interface diffuse across the entire interface and penetrate into each other. Until the distance between atoms is small enough, new chemical bonds can form.

The terminal reactive groups of polymers A and B penetrate into each other, and the interpenetrating behavior can be simplified as a one-dimensional model, as shown in Figure 9.

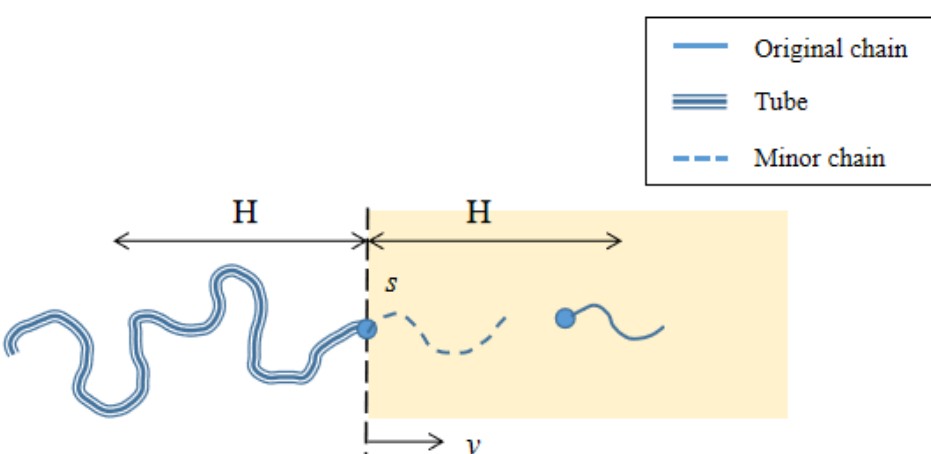

**Figure 9.** A schematic to show the diffusion behavior of the $i$th polymer chain across the interface.

In order to model the penetration of chains, the Stokes–Einstein model was used to assume the diffusion of the $i$th chain. We assumed that the polymer chains diffuse along their contour tube analogous to the motion of a snake. This curvilinear motion is characterized by the Rouse friction model. We assumed the environmental stimulus promotes the diffusion of the chain segments by $F_s$, which is the projection of force $F$ to the coordinate $s$ of the tube. We constructed two coordinate systems: $s$ denotes the curvilinear path along the minor chains, and $y$ denotes the linear coordinate perpendicular to the contact interface. In the Rouse friction model, the curvilinear diffusion coefficient of the $i$th chain is written as

$$D_i = k_B T / n_i \xi \tag{14}$$

where $\xi$ is the Rose friction coefficient per unit Kuhn segment.

The environmental stimulus promotes the diffusion of the chain segments by $F_s$, where $F_s = F < \cos\theta >$, $\theta$ is the angle of the direction of the stimulus field and the tangent direction of $s$, and $<\cos\theta>$ is the statistical average along the whole chain. The diffusion coefficient caused by $F_s$ is written as

$$D_{F_s} = F_s / \xi \tag{15}$$

The diffusion of the $i$th chain along the curvilinear coordinate $s$ is stated by

$$\frac{\partial C_i}{\partial t} = D_i \frac{\partial^2 C_i}{\partial s^2} - D_{F_s} \frac{\partial C_i}{\partial s} \tag{16}$$

where $C_i \left( mol/m^3 \right)$ is the molar concentration of the reactive groups on the $i$th chain [37,38]. The conversion of $s$ and $y$ in the two coordinate systems is expressed as

$$\frac{s}{b} = \frac{(y/b)^2}{1 + (y/b)^2 (\cos\theta/2)^2} \tag{17}$$

If we use the temperature field as the environmental stimulus, Equation (16) is reduced to $\frac{\partial C_i}{\partial t} = D_i \frac{\partial^2 C_i}{\partial s^2}$, and Equation (17) is simplified by $y = \sqrt{sb}$.

### 2.3.3. Diffusion–Reaction Model around the Interface

Due to the diffusion of polymer segments, the terminal reactive groups on the interface penetrate into each other. Then, new chemical bonds and new long chains are formed. The movement of chain segments can be understood by two processes: chain diffusion and ending group reactions.

Ending group reactions are mostly bond exchange reactions or compound reactions. The reactions can be written as $A + BE + F$ or $A + BE$.

In this article, we assumed that the rate of the molar concentration changing over time of the associative groups in both bond exchange reactions and compound reactions was identical. Therefore, we only considered the compound reaction $A + BE$ in the following.

We denoted the molar concentrations of the ending groups in the $i$th A chain and $i$th B chain as $C_{iA} \left( mol/m^3 \right)$ and $C_{iB} \left( mol/m^3 \right)$, respectively. Correspondingly, the molar concentrations of the dissociated groups in the $i$th A chain and $i$th B chain were denoted as $C_{iA}^d \left( mol/m^3 \right)$ and $C_{iB}^d \left( mol/m^3 \right)$, respectively. Additionally, the molar concentration of the associated groups in the $i$th E chain was denoted as $C_{iE}^a \left( mol/m^3 \right)$. The molar concentration of the dissociated groups was denoted as $C_i^d \left( mol/m^3 \right)$.

We took the reaction rate on the $i$th chain as $k_i$, which depends on the chain force $f_i$. At the relaxed state, the reaction rate reduces to that at the stress-free state $k_i^0$. It can be understood that there is a transition energy barrier in the chemical reactions, according to the Bell model, and the chain force $f_i$ can change the energy barrier [39].

During the compressed state, the transition energy barrier decreases. We supposed the transition energy barrier of the original reaction as $\Delta G$, and the energy barrier decreases to $\Delta G - f_i \Delta x$ because of the chain force $f_i$ when the $i$th chain is compressed, as shown in Figure 10.

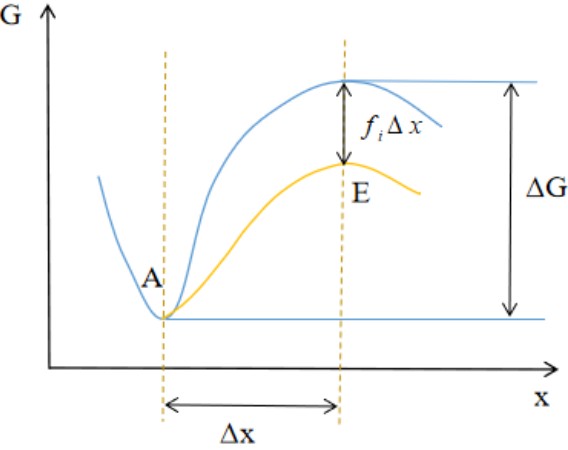

**Figure 10.** Schematic diagram to show the energy profile of the chemical reaction kinetics.

The reaction rate $k_i$ is governed by the energy barrier through exponential functions as

$$k_i = A \exp\left(-\frac{\Delta G - f_i \Delta x}{k_B T}\right) = k_i^0 \exp\left(\frac{f_i \Delta x}{k_B T}\right) \tag{18}$$

where $\Delta x$ is the distance along the energy landscape coordinate.

During the compressed state ($\lambda_1 = 1/\lambda$, $\lambda_2 = \lambda_3 = \sqrt{\lambda}$), the chain force $f_i$ of the $i$th chain is written as

$$f_i = \frac{\partial w_i}{\partial r_i} = \frac{k_B T}{b}\beta_i = \frac{k_B T}{b}L^{-1}\left(\sqrt{\frac{(2\lambda + \lambda^{-2})}{3n_i}}\right) \tag{19}$$

The chemical kinetics during the compressed state can be written as

$$\partial C_{iE}^a / \partial t = k_i C_{iA}^d C_{iB}^d \tag{20}$$

Integrating the group reactions (Equation (20)) and the chain diffusion (Equation (16)) [40], the effective diffusion–reaction model can be formulated as

$$\begin{cases} \frac{\partial C_{iA}^d}{\partial t} = \frac{k_B T}{n_i \zeta}\frac{\partial^2 C_{iA}^d}{\partial s^2} - \frac{F_s}{\zeta}\frac{\partial C_{iA}^d}{\partial s} - \frac{\partial C_{iE}^a}{\partial t} \\ \frac{\partial C_{iB}^d}{\partial t} = \frac{k_B T}{n_i \zeta}\frac{\partial^2 C_{iB}^d}{\partial s^2} - \frac{F_s}{\zeta}\frac{\partial C_{iB}^d}{\partial s} - \frac{\partial C_{iE}^a}{\partial t} \\ \partial C_{iE}^a / \partial t = k_i C_{iA}^d C_{iB}^d \end{cases} \tag{21}$$

We assumed the diffusion and reactions only occur in the interpenetrating region which is close to the contact interface. The width of this region is 2H with two boundaries as $y = \pm H$, and the boundary of the corresponding $s$ coordinate is $s = \pm s_H$.

Next, the boundary conditions of Equation (21) are described as

$$C_{iA}^d(s = -s_H, t) = 0, C_{iB}^d(s = s_H, t) = 0 \tag{22}$$

$$C_{iE}^a(s, t = 0) = 0 \tag{23}$$

$$C_{iA}^d(s, t = 0) = N_{iA}\Psi(s), C_{iB}^d(s, t = 0) = N_{iB}(1 - \Psi(s)) \tag{24}$$

where $\psi(s)$ is defined as $\psi(s) = 1$ at $s > 0$, $\psi(s) = 0.5$ at $s = 0$, and $\psi(s) = 0$ at $s < 0$.

Integrating Equations (18) and (20), the molar concentration of the associated groups is

$$C_{iE}^a(s, t) = \int_0^t k_i C_{iA}^d(s, \tau) C_{iB}^d(s, \tau) d\tau \tag{25}$$

The effective volume density of the $i$th new chain in the interpenetrating region is

$$N_{iE}^{ha}(t) = \left(\int_{-s_H}^{s_H} C_{iE}^a(s, t)ds\right)/s_H \tag{26}$$

The free energy density in the interpenetrating region can be written as

$$W^{ha} = \sum_{i=1}^{m} N_{iE}^{ha} n_i k_B T[\beta_i / \tanh\beta_i + \ln(\beta_i / \sinh\beta_i)] \tag{27}$$

where $\beta_i = L^{-1}\left(\sqrt{[(2\lambda + \lambda^{-2})]/3n_i}\right)$.

The Cauchy stress along the $\lambda_1$ direction can be written as

$$\sigma_{12}^{ha} = \lambda_1 \frac{\partial W^{ha}}{\partial \lambda_1} = k_B T \frac{(1 - \lambda^{-3})}{\lambda\sqrt{6\lambda + 3\lambda^{-2}}}\sum_{i=1}^{m} N_{iE}^{ha}\sqrt{n_i}L^{-1}(\sqrt{\frac{2\lambda + \lambda^{-2}}{3n_i}}) \tag{28}$$

There are zone A, the interpenetrating zone and zone B in the networks of polymers after interface bonding is formed. In summary, after the environmental stimulus is applied, the model of the interpenetrating region, which forms at the interface between polymers A and B, is established. The relationship between the interface bonding strength and time is also shown in Equations (25)–(28).

*2.4. The Relationship between Interpenetration around the Interface and the Actual Contact Area Ratio r*

Okada, H. et al. proposed that the actual bonding area is basically identical to the actual contact area in the field of seal bonding [41]. Therefore, we can use the actual contact area to represent the actual bonding area.

In the compression experiments of Pang, M.H. et al., the process of liquid infiltration between the modified sealed surfaces was observed in situ [34]. Firstly, initial compressive stress $\sigma_0$ was applied in sealing pairs, and the initial actual contact area ratio $r_0$ could be obtained. The compressive stress and the actual contact area ratio $r$ at different times could be determined by the stress relaxation characteristic curve of rubber elastomers and the in situ compression experiments, respectively. Secondly, once an environmental stimulus was introduced, the change in the actual contact area ratio $r$ could be obtained. Then, after the environmental stimulus was removed, the bonding strength $\sigma_{12}^{hq}$ (MPa) could be characterized from tensile tests.

Due to the existence of diffusion and bonding reactions during the interface bonding processes, the distance between atoms around the interface is reduced. Therefore, the value of the actual contact area ratio $r$ after interface bonding is bigger than that before bonding.

**3. Results**

In view of the disadvantages of the current static sealing methods, a new sealing method is proposed in this study, named alterable static sealing, which is suitable for some complex and difficult sealing requirements.

The influence of the material modulus, the interface roughness, the contact interface stress, etc., on the sealing performance has been widely studied in the existing mathematical models of pre-compression static sealing. In alterable static sealing, interface bonding is another important factor that affects the sealing performance. We used the actual contact area ratio $r$ to characterize the sealing performance. Considering the influence of interface bonding, the mathematical model of pre-compressing static sealing was revised to obtain the mathematical model of alterable static sealing.

In the field of alterable static sealing, the sealed surfaces are always compressed. The relationship between stress and the actual contact area ratio $r$ is shown in Equation (3). The corrective results of the actual contact area ratio $r$, which include the influence of interface bonding, were obtained from this study.

We took the time that the seals work under the pre-compressing static sealing state before applying the environmental stimulus as $t_p$. When the environmental stimulus was applied, we took the actual contact area ratio as $r_p$.

When the environmental stimulus was applied, the actual contact area ratio $r_p$ caused by the compressive stress and stress relaxation of polymers was regarded as the initial

condition for the formation of interface bonding. Therefore, the sealing performance of alterable static sealing could be shown as

$$
r = \begin{cases}
\left( \dfrac{\varepsilon_0 E_n + \varepsilon_0 \sum\limits_{i=1}^{n-1} E_i e^{-t/\tau_i}}{A_r} \right)^{1/B_r} & t < t_p \\[2em]
r_p & t = t_p \\[2em]
\left[ \left( \dfrac{\varepsilon_0 E_n + \varepsilon_0 \sum\limits_{i=1}^{n-1} E_i e^{-t/\tau_i}}{A_r} \right) + \left( \dfrac{k_B T \dfrac{(1-\lambda^{-3})}{\lambda\sqrt{6\lambda+3\lambda-2}} \sum_{i=1}^{m} N_{iE}^{ha} \sqrt{n_i} L^{-1}\left(\sqrt{\dfrac{2\lambda+\lambda^{-2}}{3n_i}}\right)}{A_r} \right) \right]^{1/B_r} & t > t_p
\end{cases}
\tag{29}
$$

Furthermore, in order to vividly describe the mathematical model of alterable static sealing, the curve between the stress relaxation of rubber elastomers, the bonding strength of the interface, the sealing performance and time $t$ was drawn, as shown in Figure 11.

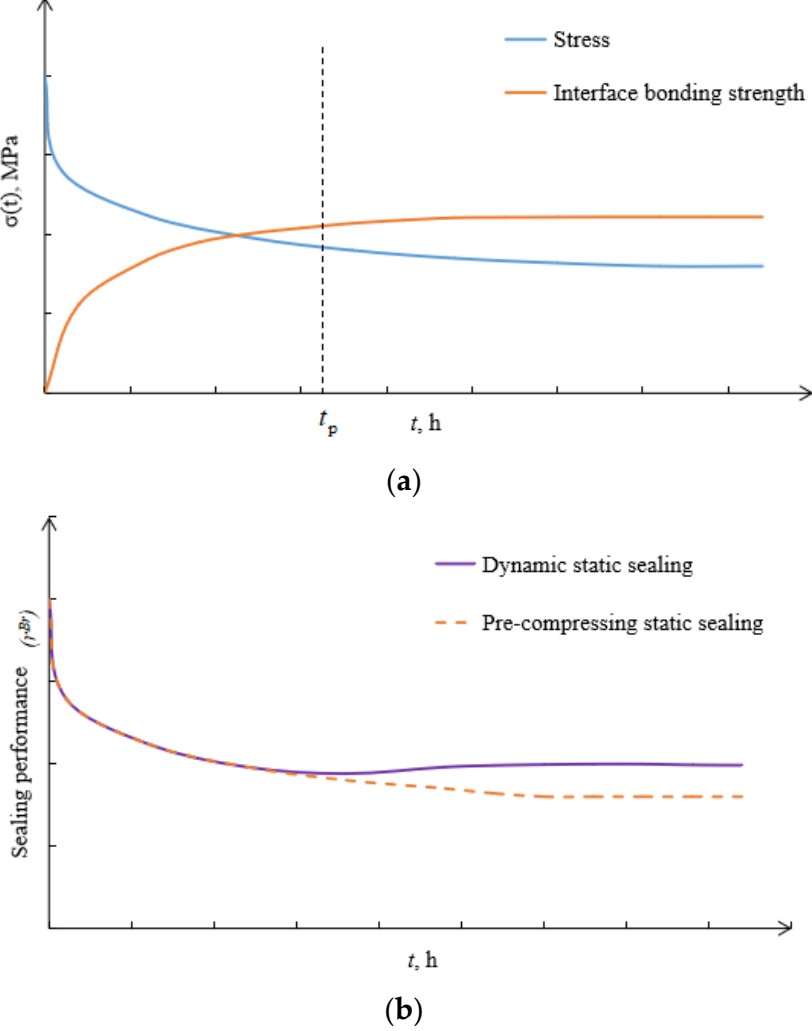

**Figure 11.** Schematics of the mathematical model of alterable static sealing: (**a**) the stress relaxation behavior (blue) and the interface bonding strength of the interpenetration region around the interface (orange); (**b**) sealing performance ($r^{B_r}$) of alterable static sealing (purple) and pre-compressing static sealing (orange).

Figure 11 shows the influence of interface bonding on the sealing performance. The diagrams in Figure 11 are the results of simulations. In Figure 11a, the stress relaxation

behavior of rubber elastomers, which shows the contact stress of the sealed interface in the field of pre-compression static sealing, is shown by the blue curve. Additionally, the change in the interface bonding strength over time is shown by the orange curve. The diagrams in Figure 11b characterize the sealing ability. In Figure 11b, the relationships between the sealing performance (represented by $r^{B_r}$) and time $t$ of pre-compressed static sealing and alterable static sealing are shown by the orange curve and the purple curve, respectively. Due to interface bonding, interface bonding strength could be formed and gradually tended to a stable value after the environmental stimulus was applied at $t_p$ (the time shown by the dotted line in Figure 11a). A part of the actual contact changed from releasable contact to irrevocable adhesive contact in the compressed state, due to the formation of interface bonding. The reduction in the actual contact area caused by the stress relaxation of rubber elastomers was compensated by this irrevocable adhesive contact; thereby, the inevitable attenuation of the actual contact area caused by the viscoelastic characteristics of rubber elastomers was partially compensated. The attenuation process of the actual contact area ratio $r$ to the threshold $r_c$ was slowed down, and the sealing performance was improved.

We compared the theoretical results of Equation (28) to the experimental results of interface adhesion in [42], and the influence on the sealing performance (Equation (29)) could be deduced. The following provides a brief description of the bases for this comparison:

(1) The effect of bonding on the sealing performance is proposed, and there are some previous bonding theories [43]. In the sealing formed by the sealant and joint sealant, the sealing force is increased as the bonding between contact surfaces is achieved, and a stable bonding strength is formed. Meanwhile, the bonding area becomes the actual contact area [41,44]. The probability of leakage is reduced, and the sealing performance is improved.

(2) According to the research of Persson, B.N.J. et al. [3,31–33], the leakage rate could be represented by the actual contact area ratio $r$, in order to represent the sealability. As long as the value of the actual contact area ratio $r$ is bigger than the threshold $r_c$, leakage channels can almost be ignored.

(3) According to the research of Pang, M.H. et al. [34], the sealing force and $r$ change in a power function law, and the exponent of the power function is $B^r$.

Next, we used $r^{B_r}$ to indicate the sealing performance in Equation (29). According to Pang, M.H. et al., $r^{B_r}$ is positively correlated with the sealing force. According to Persson, B.N.J. et al., as long as $r^{B_r} > r_c^{B_r}$, the sealing performance is good. The interfacial bonding in our sealing method depends on thermal motions and chemical reactions of polymers which are stimulated by an environmental stimulus. There has previously been experimental research of interface bonding processes [4]. The interfacial bonding force between two sealed surfaces will increase the value of the sealing force, which will increase the actual contact area ratio $r$ and improve the sealing performance.

In [42], CuCl2-loaded polybutadiene rubber was employed. Two pieces of rubber were bonded under the manipulation of ultraviolet rays (40 mW/cm$^2$) because disulfide metathesis occurred. The stress–strain curves of specimens under different bonding times are shown in Figure 12.

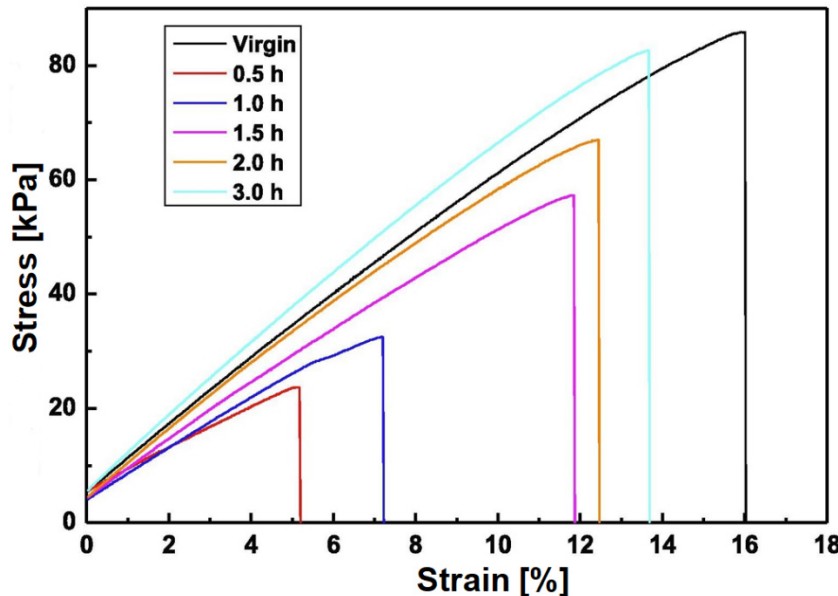

**Figure 12.** Schematics of the stress–strain curves of specimens under different bonding times [42].

It can be concluded that the relationship between interface stress and the bonding time calculated in this research shows good agreement with the experimental results in [42], as shown in Figure 13. The used parameters can be found in Table 1. The chain dynamics parameters and Rouse friction coefficients were within the reasonable order compared with the results in [38,45].

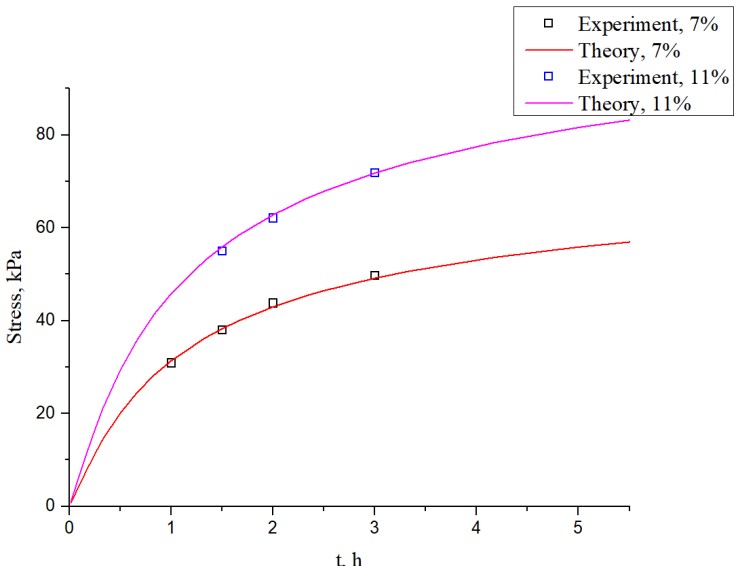

**Figure 13.** Schematics of the relationship between interface stress and bonding time in theoretical calculation and experiments.

**Table 1.** Model parameters used in this paper.

| Parameter | Definition | Simulation Data |
|---|---|---|
| $k_i^0(s^{-1})$ | Reaction rate | $1.7 \times 10^{-4}$ |
| $\Delta x(m)$ | Distance along the energy landscape coordinate | $4 \times 10^{-9}$ |
| $b(m)$ | Kuhn segment length | $5.2 \times 10^{-10}$ |
| $\xi(N/m)$ | Rouse friction coefficient | $2 \times 10^{-3}$ |

Figure 13 shows both the theoretical simulation values and experimental results of the interface stress at two different deformation ratios (7% and 11%). As the bonding time increased, the interface stress gradually increased, and the growth rate gradually slowed down. The experimental values and the theoretical values show good agreement overall. Because some necessary simplifications have been made in the theoretical model, and the effects of chain entanglement and chain crosslinking were ignored, there were some deviations. When the bonding time is longer (such as 3 h in this example), the diffusion of segments at the interface is deeper, and there are more reactions of the terminal reactive groups. Both of them promote the degree of chain entanglement between the surfaces and the increase in chain co-crosslinking. Therefore, under a longer bonding time, the theoretical simulation values are slightly lower than the experimental results.

Due to the fact that the seals are always compressed in static sealing, bonding failures caused by cohesive failure and interface tearing can be ignored.

## 4. Conclusions

In this research, aiming at the disadvantages of the two existing representative static sealing methods, a new sealing method, alterable static sealing, was proposed. The actual contact area ratio *r* was used to characterize the leakage rate, which is a direct standard for sealability. Based on the diffusion–reaction equation of the interfacial interpenetrating network, the interface bonding strength $\sigma_{12}^{ha}$ was obtained, and the influence of interface bonding on pre-compressing static sealing was described quantitatively. We revised the model of pre-compressing static sealing by considering the interface bonding process, and the model of alterable static sealing was established at the same time. We proved that the sealing performance could be quantitatively improved in alterable static sealing based on this model. Moreover, the characteristics and a scheme of the implemented sealing processes of alterable static sealing were proposed.

Laboratory tests of our new concepts are very necessary. The difficulty, workload and complexity of the experimental design and tests are large, and we are trying to conduct the experimental part which will be published in our other contribution. In the experimental part, we will manufacture samples which are sealed with pre-compressing static sealing and our new static sealing method and compare the difference in their sealing performances. The theoretical model and the laboratory tests have a certain degree of independence. First, in this research, we described the theoretical model.

In alterable static sealing, the advantages of pre-compression static sealing are maintained, such as simple processes, and convenience for disassembly. However, the dependence on the compression deformation capacity and the requirement for the relaxation characteristics of rubber elastomers are reduced. As the actual contact area could be compensated by interface bonding, sealed surfaces with lower dimensional accuracy can be used as well. The findings presented in this paper could provide meaningful guidance for static sealing, especially under complex environments, for example, static sealing in PEMFCs and RFBs.

**Author Contributions:** Conceptualization, Z.W. and X.H.; methodology, Y.W.; investigation, J.Z.; writing—original draft preparation, X.H.; formal analysis, Y.K. All authors have read and agreed to the published version of the manuscript.

**Funding:** This research was funded by National Key R&D Program of China (grant number 2020YFB0106600) and Tsinghua University-Weichai Joint Research Institute of Power and Intelligent Manufacturing (grant number WCDL-GH-2020-0215). And the APC was funded by Department of Mechanical Engineering, Tsinghua University.

**Institutional Review Board Statement:** Not applicable.

**Informed Consent Statement:** Not applicable.

**Data Availability Statement:** The study did not report any data. Data sharing not applicable.

**Acknowledgments:** This work was supported by the National Key R&D Program of China (No. 2020YFB0106600), Tsinghua University-Weichai Joint Research Institute of Power and Intelligent Manufacturing (No WCDL-GH-2020-0215), Tsinghua University (Department of Mechanical Engineering)-Anhui Zhongding Sealing Parts CO., Ltd. Joint Research Center for Rubber and Plastic Seals, Tsinghua University Initiative Scientific Research Program. All individuals included in this section have consented to the acknowledgement.

**Conflicts of Interest:** Yuchao Ke is the employee of Key Laboratory of High-Performance Rubber & Products of Anhui Province. The paper reflects the views of the scientists, and not the company.

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
