# Peer review of "Study on a New Static Sealing Method and Sealing Performance Evaluation Model for PEMFC"

_wevj, doi:10.3390/wevj12040237_

Round 1

Reviewer 1 Report

The authors proposed a new concept of joining and sealing joints. It was most likely created on the basis of sealing problems in proton exchange membrane fuel cells (PEMFCs). This concept is interesting, but the authors presented basically the idea itself and a certain mathematical description of the situation only. I have the impression that the rest of the article is missing. Chapter 4 (results) presents the diagrams in Figure 11. Are they the result of simulations obtained from equation (28), or only a schematic representation of the expected course? Have the authors carried out any laboratory tests of seals made according to the new concept? Some test results are needed. The authors mention the stimulation that causes the seal formation. Could it be just higher temperature or other factors? More details are needed. It is assumed that the reader may repeat the same research on the basis of the article.
The name of the new method "dynamic static sealing" does not seem to be the best. It contains two mutually exclusive words: dynamic and static. What is the justification for such a name?

In conclusion, it seems to me that some parts of the manuscript have been expanded while others are in a residual state. The experimental results and possible discussion of the results are missing. Hence, the conclusions are meager. What is presently in the manuscript does not correspond to the required structure of a typical article. It seems to me that the authors should try to complete the missing parts of the manuscript.

Author Response

Thank you for your nice comments.

These comments are all valuable and very helpful for revising and improving our paper, as well as the important guiding significance to our researches. We have studied comments carefully and have made corrections which we hope meet with approval.

The responds to your comments are shown in the attachment.

Reviewer 2 Report

The manuscript reported the characteristics and shortcomings of the current sealing methods for PEMFCs. At present, the deterioration of sealing performance was an important factor that limited the service life of PEMFCs. Hence, a novel approach to static sealing was proposed and named dynamic static sealing. In addition, the actual contact area ratio r was used as the standard for sealability. Based on the mathematical model of pre-compressing static sealing, the influence of interface bonding was considered, and the mathematical model of dynamic static sealing was established.

I consider the content of this manuscript will definitely meet the reading interests of the readers of the WEVJ journal. However, there are certain English spelling and grammar issues, and also the discussion and explanation should be further improved. In particular, the proposed novel sealing technology may be applicable to other battery systems of electric vehicles, not only limited to PEMFCs.

Therefore, I suggest giving a minor revision and the authors need to clarify some issues or supply some more experimental data to enrich the content. This could be a comprehensive work after revision.

My detailed comments can be found in a separate PDF file.

Author Response

(The authors gave the same response as above.)

Round 2

Reviewer 1 Report

I think that the changes made by the authors following comments from reviewers have improved the manuscript so that it can now be published.

Reviewer 2 Report

I have carefully read the author's revised version, and I thank the author for his careful consideration and careful revision of my previous comments. Although the previous of my comments are minor revisions, the authors treat them seriously and huge efforts have been made to improve the manuscript. 

I am satisfied with the current version and think it is acceptable. I will not make any other modification suggestions.